# Corticosteroids for COVID-19 Therapy: Potential Implications on Tuberculosis

**DOI:** 10.3390/ijms22073773

**Published:** 2021-04-06

**Authors:** Radha Gopalaswamy, Selvakumar Subbian

**Affiliations:** 1Department of Bacteriology, ICMR-National Institute for Research in Tuberculosis, Chennai 600031, India; radhagopalaswamy@gmail.com; 2Public Health Research Institute at New Jersey Medical School, Rutgers University, 225 Warren Street, Newark, NJ 08854, USA

**Keywords:** SARS-CoV-2, coinfection, immunosuppression, latency, *Mycobacterium tuberculosis*, reactivation, antibodies, inflammation

## Abstract

On 11 March 2020, the World Health Organization announced the Corona Virus Disease-2019 (COVID-19) as a global pandemic, which originated in China. At the host level, COVID-19, caused by the Severe Acute Respiratory Syndrome Coronavirus-2 (SARS-CoV-2), affects the respiratory system, with the clinical symptoms ranging from mild to severe or critical illness that often requires hospitalization and oxygen support. There is no specific therapy for COVID-19, as is the case for any common viral disease except drugs to reduce the viral load and alleviate the inflammatory symptoms. Tuberculosis (TB), an infectious disease caused by *Mycobacterium tuberculosis* (Mtb), also primarily affects the lungs and has clinical signs similar to pulmonary SARS-CoV-2 infection. Active TB is a leading killer among infectious diseases and adds to the burden of the COVID-19 pandemic worldwide. In immunocompetent individuals, primary Mtb infection can also lead to a non-progressive, asymptomatic latency. However, latent Mtb infection (LTBI) can reactivate symptomatic TB disease upon host immune-suppressing conditions. Importantly, the diagnosis and treatment of TB are hampered and admixed with COVID-19 control measures. The US-Center for Disease Control (US-CDC) recommends using antiviral drugs, Remdesivir or corticosteroid (CST), such as dexamethasone either alone or in-combination with specific recommendations for COVID-19 patients requiring hospitalization or oxygen support. However, CSTs can cause immunosuppression, besides their anti-inflammatory properties. The altered host immunity during COVID-19, combined with CST therapy, poses a significant risk for new secondary infections and/or reactivation of existing quiescent infections, such as LTBI. This review highlights CST therapy recommendations for COVID-19, various types and mechanisms of action of CSTs, the deadly combination of two respiratory infectious diseases COVID-19 and TB. It also discusses the importance of screening for LTBI to prevent TB reactivation during corticosteroid therapy for COVID-19.

## 1. Introduction

The highly contagious COVID-19 is an ongoing global pandemic. There are about 116 million COVID-19 cases globally, with nearly 2.5 million deaths reported until 8 March 2021 [1]. Although COVID-19 primarily affects the respiratory system, including the lungs, the virus and disease can disseminate to other organs rapidly [2,3]. Following aerosol exposure, SARS-CoV-2, the causative agent of COVID-19, usually progresses into the upper and conducting airway from the nasal cavity and can develop infection symptoms, including rhinitis, sore throat, and sneezing, or remain asymptomatic, depending on the host innate immune response. COVID-19 is diagnosed by several molecular and immunological methods across the world [4]. Based on the clinical presentation, COVID-19 is classified into an initial phase with mild-to-moderate disease symptoms, such as cough, fever, pneumonia, and chest radiology (computed tomography or CT) impressions prominently reveal “ground-glass” appearance of lungs with infiltrates. In the second, severe disease phase, COVID-19 cases show symptoms including oxygen saturation deficit with a spatial oxygen pressure (SpO2) below 92%. The critical phase of COVID-19 involves acute respiratory distress syndrome (ARDS) associated with shock, myocarditis, heart failure, kidney failure, liver injury, and death [5,6,7,8]. However, the molecular events underpinning the disease pathology at different stages of COVID-19 are not fully understood.

Patients with severe and critical COVID-19 have high systemic levels of inflammatory molecules, predominantly interleukins (IL-2, IL-6, IL-10), interferon- γ (IFN-γ) inducible protein 10 (IP10), monocyte chemoattractant protein 1 (MCP-1), granulocyte-macrophage colony-stimulating factor (GM-CSF) and tumor necrosis factor-alpha (TNF-α) along with lymphopenia. Besides, elevated immune cell infiltrations in the lungs lead to severe inflammation, cellular immune response failure, and the onset of a “cytokine storm” in the severe COVID-19 patients [6,7,9]. Inflammatory cytokines, such as IL-6, IL-1β, and IFN-γ that are elevated in the biospecimen of COVID-19 patients, can activate the Janus kinase (JAK)/signal transducer of activators of transcription (STAT) JAK/STAT pathway and induce the NF-kB signaling (Figure 1). Subsequent nuclear translocation of NF-KB and p38 phosphorylation leads to induction of inflammatory cytokines and chemokines responsible for the “cytokine storm”. Therefore, the immunomodulation of NF-kB activation and p38 MAPK have been considered a possible therapeutic target for severe COVID-19 cases [10,11,12]. Besides, transcriptome analysis has shown that induction of the activator protein 1 (AP-1) pathway is crucial as it connects the MAPK pathway with T-cell function, associated with milder inflammation during COVID-19 [13]. Taken together, these studies implicate inflammation as the major driving event in COVID-19 disease pathology.

Thus, one of the primary objectives in treating COVID-19 is to alleviate inflammation by treating with anti-inflammatory drugs, such as corticosteroids (CST), including dexamethasone, prednisone, methylprednisone and hydrocortisone. Among the CSTs, dexamethasone, administered at 6 mg daily dose (oral or IV), is the preferred choice to treat severe COVID-19 cases. During non-availability of dexamethasone, the United States–National Institute of Health (US-NIH) recommends using other CSTs at equivalent dose, including prednisone 40 mg; methylprednisone 32 mg, and hydrocortisone 160 mg. The half-life, period of activity, and administration frequency should be carefully considered where alternate CSTs are used. (Table 1) [14,15,16].

In addition, although there is no “bonafide” treatment available, the US-NIH and the Center for Disease Control (US-CDC) recommend using Remdesivir, CST and anti-SARS-CoV-2 monoclonal antibodies to treat COVID-19 patients at various stages of disease severity. The anti-SARS-CoV-2 antibodies, Bamlanivimab and Etesevimab, are currently recommended for emergency use during the early stages of COVID-19 when no hospitalization is required [14,15,16]. Still, the risk of progression to severe disease is high with this treatment. Remdesivir, an antiviral drug, is approved by the Federal Drug Administration (US-FDA) to treat patients with a high -risk of disease progression, mostly among hospitalized individuals who do not require oxygen support. For COVID-19 cases requiring hospitalization and oxygen support, CST therapy using dexamethasone is recommended either alone or in combination with Remdesivir. For these patients, Tocilizumab is currently recommended in conjunction with dexamethasone therapy [14,15,16].

Although CSTs have anti-inflammatory activities, which are beneficial to dampen inflammation caused by infection, such as SARS-CoV-2, CST usage should be considered very carefully due to their immunosuppressive properties, particularly in individuals with underlying health conditions. In such cases, CST administration may not only exacerbate the disease symptoms, but it can also elevate the risk of opportunistic secondary infections, such as pneumonia or worsening of the existing pulmonary infectious disease, like TB. Importantly, in most hospitalized patients with respiratory infectious diseases, a sudden cough or chest radiology changes would be confused as secondary bacterial pneumonia due to nosocomial infection rather than TB in the first instance by a clinician. This situation would delay the diagnosis of TB and probably increase the number of undiagnosed cases over a period contributing to the overall global TB burden.

TB, a pulmonary disease caused by *Mycobacterium tuberculosis* (Mtb), is a leading killer among infectious diseases of humans [17,18]. Similar to SARS-CoV-2, Mtb transmission occurs through aerosol droplets generated by individuals with active disease, although SARS-CoV-2 can be transmitted from asymptomatic cases [19]. The pathophysiological spectrum that ensues Mtb exposure ranges from complete elimination of infection (a person with prior exposure immediately clears the bacilli by innate or acquired immunity) to symptomatic active TB disease to the establishment of LTBI, in which the disease progression is not immediate [20,21]. However, LTBI cases can reactivate to symptomatic, active TB upon immune compromising health conditions, including treatment with anti-TNF*α* antibodies or CSTs [22,23,24,25].

At present, COVID-19 cases are not screened for active TB or LTBI before starting CST therapy, either alone or adjunct to antiviral agents. Since CST possesses host immune suppression properties, treatment with CST poses a significant risk for acquiring nosocomial infections and reactivation of LTBI or exacerbating existing TB among COVID-19 cases. To determine if CSTs can be used as a host-directed therapy for COVID-19, it is crucial to understand their mechanism of action for both anti-inflammatory and immunosuppressive functions. Similarly, the impact of CST therapy in patients co-infected with both SARS-CoV-2 and Mtb needs to be delineated. In this review, we summarize the cellular signaling underpinning CST activities, discuss the therapeutic use of CSTs for COVID-19 and highlight the implications of CST therapy on active and latent TB.

## 2. General Therapeutic Applications of Corticosteroids

CSTs are either natural hormones or synthetic compounds and function as critical regulators of whole-body homeostasis to help an organism resist environmental changes and foreign agents’ invasion. Functionally, CSTs are broadly classified into glucocorticoid (intermediary metabolism, inflammation, immunity, wound healing, myocardial, and muscle integrity) and mineralocorticoid (salt, water, and mineral metabolism). Most CSTs possess overlapping glucocorticoid or mineralocorticoid functions [26]. CSTs are prescribed in varying doses and duration for various human diseases like inflammatory disorders of infectious and non-infectious etiologies and allergies (Table 2). Therapeutically, CSTs are divided into three categories the oral/intravenous (IV) or the inhaled and the topical forms [27,28]. The types of corticosteroids, route of administration, and their therapeutic use in different clinical conditions are highlighted in Table 2 [27,28].

## 3. Mechanism of Action of CST

CSTs can cross the host cell membrane and interact with the glucocorticoid receptors (GRs); the GR-CST complex translocate into the nucleus and binds to glucocorticoid response elements (GRE), which regulates the expression of several genes (Figure 1). The alteration of gene transcription through GRE exerts anti-inflammatory activities. CSTs inhibit the production of proinflammatory mediators, including phospholipase A2, in immune cells such as macrophages, eosinophils, lymphocytes, dendritic cells, and mast cells [28]. Furthermore, CST dampens host cell functions associated with inflammation, including immune cell extravasation, epithelial cell adhesion, chemotaxis, phagocytosis, and production of antimicrobial effector molecules by the immune cells [29]. At the molecular level, CSTs exert their effects by transactivation and transrepression pathways and induce anti-inflammatory genes, like the glucocorticoid-induced leucine zipper (GILZ) and dual-specificity phosphatase (DUSP1) [30]. In T-cells and macrophages, GILZ interacts with regulatory proteins, such as NF-kB, AP-1, Ras1, C/EBP, to control proinflammatory gene expression [31,32]. Specifically, GILZ prevents NF-kB activation by directly binding to the p65 subunit of NF-kB. At the same time, it inhibits AP-1 by interacting directly with c-Fos and c-Jun, preventing DNA binding, and interacting with Raf-1, inhibiting its phosphorylation and subsequent activation of mitogen-activated protein kinase kinase (MAPKK), extracellular signal-regulated kinase 1/2 (ERK1/2), and AP-1 [33]. Similarly, DUSP-1 inactivates p38 kinase-mediated proinflammatory signaling in immune cells. The p38 MAPK pathway is induced mainly by proinflammatory factors and environmental stresses, which prominently impact a subset of physiological events, including inflammation [32]. Additionally, CSTs such as dexamethasone can induce the production of host cytochrome P450-3A4, an enzyme capable of reducing the effectiveness of other concomitant medications used for treatment [33,34,35].

CSTs have been shown to upregulate certain anti-inflammatory molecules, including IL-10 and CD163, by inducing annexin A1 (also called lipocortin 1)-mediated pathway in macrophages and monocytes [34,35]. Annexin A1 (encoded by ANXA1) binds to its receptor ALXR and dampens phospholipase A2-mediated eicosanoid production, which is a crucial anti-inflammatory mechanism. CSTs activate both ANXA1 and ALXR and exploit the juxtacrine interaction of innate cells expressing ANXA1 and ALXR to mediate the anti-inflammatory effect [36]. The anti-inflammatory IL-10 represses proinflammatory cytokines while inducing the expression of other anti-inflammatory mediators like GILZ, DUSP, IL-1RA and CD163 [34]. CD163 is a scavenger receptor of monocyte-macrophage lineage and proven to be upregulated in response to CSTs treatment. CD163 serves as a receptor for the haptoglobin-hemoglobin complex and mediates its clearance which resolves inflammation. Recent studies demonstrated the use of CD163-targeted dexamethasone-loaded liposomes as a novel inflammatory therapy [34,37].

## 4. Corticosteroid Therapy for COVID-19

In the Randomized Evaluation of COVID-19 Therapy (RECOVERY) trial conducted by researchers at Oxford University, oral or inhaled dexamethasone administration showed lower mortality among COVID-19 patients in mechanical ventilation or oxygen support compared to those receiving no respiratory support. However, many patients in this study had comorbid conditions, such as diabetes, heart disease, TB, and human immunodeficiency virus (HIV) disease. This study had prompted the National Institute of Health (NIH) to recommend using 6 mg dexamethasone once-daily dose only to treat severely ill COVID-19 cases. However, the biologically effective dose of other CSTs remains unknown [14,38]. The benefits of CST as adjuvant therapy have been reiterated by several independent research that favored its use for COVID-19 patients with ARDS. The administration of methylprednisolone, dexamethasone, hydrocortisone, and prednisone either orally or intravenously was reported to lower the death rate of COVID-19 patients with ARDS [39]. Reports of three different trials and metanalysis showed the beneficial effects of CST therapy in treating ARDS-associated with COVID-19 patients. However, the analysis indicated a bias due to routine CST use in the usual care group, and lack of their elimination during screening should be considered [40]. Studies show the beneficial effects of smaller doses of systemic CSTs administration for a short duration or pulse CST therapy; however, other confounders should be considered before starting the treatment [41,42]. Notably, the pooled estimates of CST therapy benefits were similar between COVID-19 and non-COVID-19 associated ARDS, and there is more than 15% reduced mortality [43].

A recent prospective metanalysis compared seven clinical trials (DEXA-COVID 19, CoDEX, RECOVERY, CAPE COVID, COVID STEROID, REMAP-CAP, Steroids-SARI) conducted to evaluate the usefulness of CST therapy for severe or critically ill patients across five continents [44]. This analysis included 1703 patients (age range 5–68 years), of which 678 were randomized to receive CST therapy, including low and high dose dexamethasone, low-dose hydrocortisone, and high-dose methylprednisolone either orally or intravenously (intermittent or continuous), and compared the clinical outcome to placebo. Results from these trials conclusively offered promise to CST therapy in reducing the mortality among COVID-19 patients that required mechanical ventilation and hospitalization [44].

Prolonged therapy or increasing CST dose did not improve viral clearance in COVID-19 patients with severe or critically ill symptoms [45]. Thus, it appears that CST can be beneficial to COVID-19 cases only within an appropriate therapeutic window period. A multi-centric study in China identified a therapeutic window using radiographic progression and serum lactate dehydrogenase (LDH) as a biomarker for the severity of COVID-19 [46]. Accordingly, LDH levels higher than two times the upper limit of normal at the time of COVID-19 detection could be a useful criterion to start CST therapy with a low to moderate dose for better viral clearance and prevent the requirement of mechanical ventilation [46].

Few studies indicated a weak correlation between CST and clinical improvement of COVID-19, particularly when the patients have milder or less severe symptoms and do not require intensive care unit (ICU) admission. A clinical study with a small number of subjects found no correlation between methylprednisone therapy and clinical outcome when the COVID-19 cases showed no respiratory distress [47,48]. Furthermore, no beneficial effect to intravenous administration of dexamethasone, hydrocortisone, or methylprednisolone was reported in multi-centric studies done in China, Brazil and Iran [49,50,51].

In September 2019, the WHO issued a guideline for the use of CST to treat COVID-19 cases with careful considerations. Accordingly, administration of systemic CST therapy with 6 mg of dexamethasone (oral or intravenous daily) or 50 mg of hydrocortisone (intravenous every 8 h) for 7 to 10 days is recommended to treat patients with severe and critical COVID-19 symptoms. However, there is a discouragement to use CST therapy in patients with mild or less severe COVID-19. Besides, most of the patients in this study had comorbidities, such as diabetes, heart disease, TB, and HIV. WHO has insisted on the cautious use of CSTs for people with diabetes and indicated an under-representation of individuals with active or LTBI in most trials [52]. The list of currently ongoing clinical trials to evaluate the usefulness of corticosteroids for COVID-19 is presented in Table 3 [53]. Several clinical trials are at the initial stages of enrolling subjects, while few completed studies have not disclosed their results. Thus, the clinical usefulness of CST therapy for COVID-19 is still unclear.

## 5. COVID-19 and Tuberculosis: A Double Debacle

Clinically, both COVID-19 and TB primarily affect the lungs and respiratory tract, with the infection and transmission initiating mostly through the aerosol route. Both diseases have common respiratory symptoms like fever, cough, respiratory distress and pneumonia, and disseminated disease in other organs, including the brain (Table 4) [43,54,55,56,57,58].

Upon infection of the pulmonary alveoli, Mtb is thought to be phagocytosed by innate immune cells, such as the alveolar macrophages and dendritic cells, which use pattern recognition receptors (PRRs), predominantly toll-like receptors, expressed on their surface to interact with the pathogen-associated molecular patterns (PAMPs) of the bacteria. Following Mtb engulfment, the phagocytes produce a plethora of proinflammatory cytokines, including TNF-α, IL-1β, IL-6, IL-12, IL-18, IL-23, and IFN-γ, and chemokines. Some of these molecules, including IL-6, IL-12, IL-1β and IL-18, also constitute the inflammatory cytokine storm seen in severe COVID-19 cases. In TB, the cytokine/chemokine milieu attracts more immune cells from the circulation to the infection site, forming the granuloma [59,60]. Mtb-infected phagocytes are at the center of granuloma, followed by layers of different types of immune cells, such as monocytes, epithelioid cells, foamy macrophages, multi-nucleated giant cells, and neutrophils. Lymphocytes, including T and B cells, are mostly present as a ring or cuff at the outer side of the granulomas [59,60]. On the other hand, the infecting Mtb deploys strategies to prevent the proinflammatory responses of infected phagocytes, leading to asymptomatic LTBI. During reactivation of LTBI, the lungs are damaged through cavitation or fibrocaseation of granulomas, which results in the occurrence of clinical symptoms. The mechanisms leading to the formation of these lesions in TB reactivation are not fully understood, although it might involve sudden leakage of otherwise confined, bacteria-loaded, necrotic materials in healthy tissues [61,62]. This may occur either due to the physical disruption of non-calcified granuloma or obstructed bronchioles filled with caseum. Mtb can proliferate exorbitantly within these cavitary granulomas, facilitating bacterial dissemination from the lungs to other organs [59,60]. Furthermore, phlegm expulsion (caseum) upon coughing or sneezing contributes to disseminating Mtb infection between individuals [61,62].

Both COVID-19 and TB have been linked as syndemic with other comorbidities, such as diabetes, obesity, chronic respiratory disorders, HIV infection, hypertension, cardiovascular disease, and immunosuppression due to drugs or organ failure. These comorbidities predispose to the disease severity and mortality in both TB and COVID-19. On the other hand, although COVID-19 and TB are common among the elderly population, COVID-19 is relatively uncommon in children, compared to pulmonary and extrapulmonary TB, which are prevalent in children as much as in adults [8,55,63,64,65,66,67]. In both COVID-19 and active pulmonary TB, the clinical manifestations of disease vary depending on the nature and infectious dose of infecting organism, the host-pathogen interactions, and the immune status of the individual. Recently, several case reports highlight the co-existence of COVID-19 and TB, and one of these respiratory illnesses was predisposing the onset or worsening the clinical outcome of the other.

The combined impact of COVID-19 and TB on patient mortality/morbidity and disease management is confounded by several factors, primarily the lack of a strategy to diagnose the cases for both diseases. At present, the diagnostic tests for TB and COVID-19 are mostly performed sequentially rather than concurrently, depending on the initial disease symptoms and prior history of exposure to Mtb. In general, tuberculin skin test (TST), interferon-gamma release assay (IGRA) and/or chest X-ray are used to determine the Mtb exposure status. However, such screening procedures are complicated by the ongoing COVID-19 pandemic. For example, a mobile X-ray unit used for TB screening was found to be a cause of nosocomial SARS-CoV-2 infection. In a case report, a patient with COVID-19 developed bloody sputum along with fever and cough. Upon testing, it was confirmed that the patient had LTBI, which progressed to an active TB disease in the presence of COVID-19 [68,69]. Similarly, in a clinical study involving patient cohorts from 8 countries, where the SARS-CoV-2 infection was nosocomial, a 10.6% fatality rate was reported among 8 out of 69 confirmed cases of COVID-19 and TB [70]. In another cohort study that evaluated the consequence of either COVID-19 or TB presenting as sequelae to one another or co-diagnosed, a 12% fatality rate of patients was reported when TB and COVID-19 co-existed [71]. Furthermore, four cases of COVID-19 and TB were reported in a hospital, with clinical presentation indistinguishable between these two diseases. However, atypical chest radiograph findings prompted timely TB diagnosis in these cases [72]. These studies indicate that although it takes a longer time for healthy individuals with LTBI to progress into active disease, co-infection with SARS-CoV-2 accelerates the progression of TB by weakening the host immunity. Hence, when the chest radiology impressions show a complex disease pathology, combined with pneumonia or acute distress syndrome, it is important to consider the concurrent diagnosis of TB and COVID-19 and proper treatment strategy [73,74,75,76,77].

## 6. Corticosteroid Therapy and Tuberculosis

The American Thoracic Society (ATS), US-CDC, the Canadian Lung Association (CLA), and the Canadian Thoracic Society (CTS) describe that Mtb-infected individuals receiving >15 mg/day of CST for 2–4 weeks are at a higher risk of developing active TB [78,79]. Although CST treatment reduced patient mortality among all TB cases, the beneficial effect of CST therapy varies based on the organ impacted by TB.

In combination with standard anti-TB drugs, dexamethasone administered during the first 6 to 8 weeks of treatment has been shown to improve the survival of TB meningitis patients over 14 years of age [80]. Similarly, an overall reduction in relative risk of 22% and 35% has been reported for patients with meningitis and pericarditis, respectively, when corticosteroid was given as adjuvant therapy, along with standard anti-TB drugs [81]. A systematic review on the use of CST to improve the clinical outcome of TB showed a significant beneficial effect of CST therapy, mainly for TB meningitis cases. In contrast, a probable short-term beneficial effect or no effect was reported for patients with pulmonary TB and TB pericarditis, respectively. Adjuvant CST indicated symptomatic relief for TB lymphadenitis and peritonitis but not TB pleuritis [82]. With no further contraindications, dexamethasone or prednisolone administration is recommended for both children and adults as part of TB meningitis therapy for four weeks [83]. A meta-analysis on the use of steroids for abdominal TB showed that the inclusion of steroids could provide some clinical improvement in treated patients [84]. However, epidemiological data for the impact of CST therapy on LTBI cases, particularly in reactivating to symptomatic TB, is lacking. In fact, CSTs can reduce the tuberculin skin test (TST), the primary screening test for LTBI [85,86]. Therefore, the incidence and prevalence of LTBI and its reactivation to symptomatic TB among CST users could be underreported [87].

## 7. Corticosteroid Treatment for TB Patients with Comorbidities

Population-based case-control studies on the use of oral or inhaled or systemic CSTs (Beclomethasone, Deflazacort, dexamethasone, hydrocortisone, methylprednisolone, prednisolone, and triamcinolone, Budesonide, Fluticasone and Ciclesonide) primarily for the treatment of asthma or COPD showed a strong association between increased active TB cases and CST use, which could be attributed to the immunosuppressive-effect of CSTs [88,89,90,91,92]. The importance of TB screening for CST users has also been implicated by an increase in primary and reactivation TB among individuals treated dermatologically with CST [25]. A study on ICS use for respiratory diseases revealed TB reactivation in patients exposed to higher ICS doses [22]. In another case study of severe COVID-19, the use of single-dose Tocilizumab (TCZ) led to progressive, symptomatic TB. Since the patient did not report any comorbidities, the impaired immune function or immunosuppression brought about by COVID-19 and/or CST treatment was attributed to the reactivation of LTBI [93]. In another case report, a hospitalized and intubated COVID-19 patients with ARDS, chest consolidation and fever, failed to respond to dexamethasone and prompted testing of the sputum. Following confirmation of TB through smear and culture diagnosis, the patient was given anti-tuberculosis therapy, which improved the clinical outcome and patient cure [94]. However, another study that evaluated ICS’s use on LTBI reactivation in COVID-19 cases concludes that the increased risk of treated patients was not attributable to the population level; instead, this study recommends using ICS based on individual clinical presentation and severity of the disease. Hence, it was suggested to validate ICS’s risk to reactivate TB, over its therapeutic value for respiratory diseases, on a case-by-case basis [95].

The health emergency guideline for COVID-19 released by the Massachusetts General Hospital recommends documenting TB history and screening before prednisone administration (>20 mg dose). The same recommendations could be considered while using dexamethasone or other CST of choice by the clinician. For cases with no history of previous active TB or LTBI available, a T-spot test can be performed, and negative results prove them less-to-no risk to initiate CST therapy. Individuals with a positive or indeterminate T-spot and known LTBI history are considered high risk to reactivate LTBI upon CST therapy. Specifically, if/when CSTs are given for less than two weeks, these individuals are considered low-risk; however, if steroid use extends beyond two weeks, they are regarded as high risk for reactivation of LTBI. In both scenarios, a primary care provider’s follow-up monitoring is recommended after the completion of steroid therapy and clinical improvement [96].

## 8. Conclusions

Severe pneumonia and ARDS associated with severe COVID-19 cases warrant a need for CST treatment during later stages of the disease to reduce inflammation and tissue damage. However, CST given early on when the viral load is low could cause immune suppression, leading to viral-induced acute pulmonary exacerbations due to the host immune system’s failure to clear the virus. This is analogous to the treatment of COVID-19 cases with type-1 interferon antibodies, which may either be beneficial or detrimental in controlling the disease, depending on the disease severity [97]. Besides, CST therapy’s immunosuppression compounded with the weakened host immunity due to viral infection can pave the way for other secondary infections to crop up. This could lead to secondary superinfections by bacterial and fungal pathogens in a nosocomial setting among hospitalized COVID-19 cases. Similarly, the use of CST for COVID-19 has the potential risk of facilitating the emergence of other quiescent illnesses, such as the reactivation of LTBI (Figure 2). The WHO reports that the number of deaths due to TB would increase by 0.2–0.4 mln in 2021 due to poor disease management, including lack of proper diagnosis, incomplete treatment, and socio-economic factors associated with the COVID-19 pandemic [15]. Since immunosuppression and reactivation of LTBI are significant concerns for CST therapy, pre-or concurrent-screening of COVID-19 cases for LTBI and documenting the previous history of exposure to Mtb and/or pulmonary TB should be mandated in clinical trials involving CST. This strategy would aid in better management of both TB and COVID-19 cases.

Furthermore, a preventive LTBI drug therapy could be recommended for COVID-19 cases identified positive for TB in diagnostic screening. In fact, the WHO and CDC guidelines recommend TB preventive therapy (TPT) for LTBI treatment. The TPT regimen includes three to six months of treatment with anti-TB drugs, isoniazid either alone or in combination with rifampicin or rifapentine [98,99]. For risk groups, such as CST users, the CDC recommends LTBI therapy when prednisone is used at 15 mg/day for at least a month [98]. Clinical trials conducted among the high-risk group of non-COVID-19 cases showed a 27–95% reduction in the incidence of active TB for those on TPT, compared to placebo, which was considered effective [100,101]. It is intriguing to consider prophylactic treatment with anti-TB drugs in parallel to CST for patients diagnosed with severe COVID-19 and LTBI. This treatment strategy could prevent the reactivation of LTBI into symptomatic TB while keeping the beneficial effects of CST in controlling the severity of COVID-19. Whether the TB treatment is effective or even feasible for patients presenting such conditions (SARS-CoV2 infection and CST) is an important point to answer. Additional and extensive clinical studies are needed to address these comorbid conditions for effective management of both COVID and TB.

## Figures and Tables

**Figure 1 ijms-22-03773-f001:**
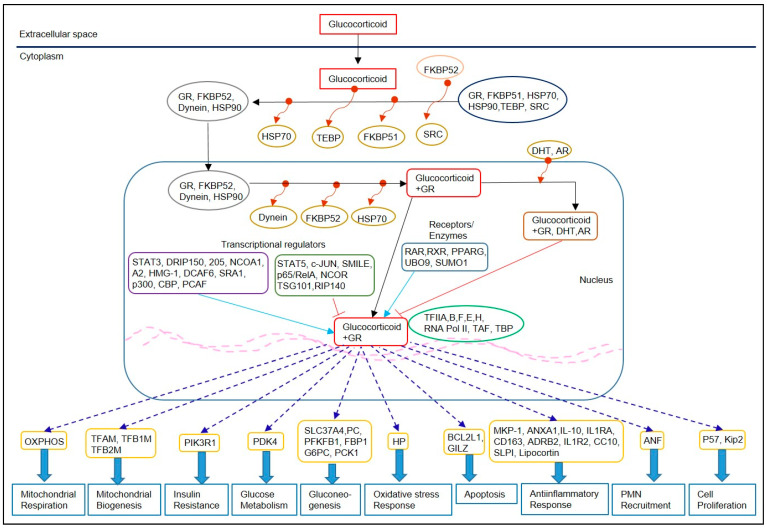
Canonical glucocorticoid receptor signaling underpinning the mechanism of action of corticosteroids. Corticosteroid (CSTs) exert their downstream effects on host cells by activating glucocorticoid receptor (GR), which modulate the transcription of several target genes. The binding of CST with GR leads to conformational changes in GR; the GR-complex translocate to the nucleus, where GR binds to GR-response elements, including positive (blue arrows) and negative (red lines) transcriptional regulators, receptors and enzymes. The downstream effects, including mitochondrial functions, metabolism, stress response, and anti-inflammatory effects of CST, are mediated through differential regulation of various effector molecules (dotted lines with arrow).

**Figure 2 ijms-22-03773-f002:**
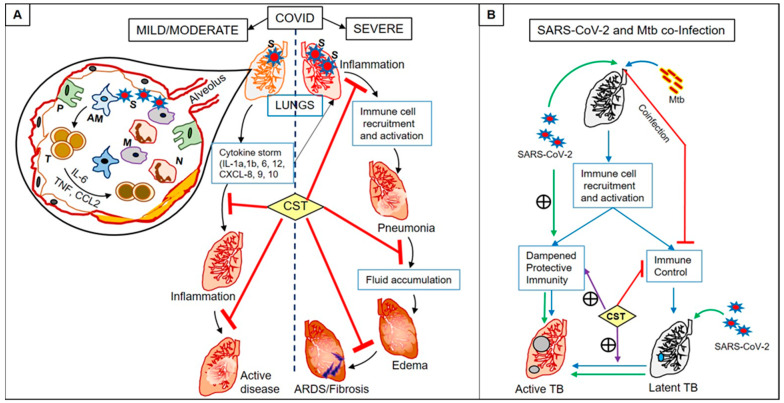
Summary of pathological events associated with Severe Acute Respiratory Syndrome Coronavirus-2 (SARS-CoV-2) and *Mycobacterium tuberculosis* (Mtb) co-infection and the effect of corticosteroid therapy. (**A**). SARS-CoV-2 enters through the pulmonary route, reaches the alveoli, and interacts with host immune cells, including alveolar macrophages (AM) and pneumocytes (P). Infection of AM produces cytokines/chemokines that recruit and activate other immune cells such as T cells (T), monocytes (M), neutrophils (N) from the blood to the infection site. Activated T cells also produce inflammatory cytokines, leading to a “cytokine storm”. In COVID-19 cases with mild and moderate symptoms, the cytokine storm-mediated inflammation is either controlled and the patient recovers; or the infection progresses to active disease with severe symptoms. The lungs of COVID-19 cases with severe symptoms showed elevated immune cell recruitment and activation, leading to pneumonia. As the disease progresses, the accumulation of fluid in the lungs results in inflammatory pulmonary edema, leading to acute respiratory distress syndrome (ARDS) and pulmonary fibrosis. Corticosteroid (CST) treatment blocks inflammation caused by the cytokine storm and prevents the formation of pneumonia, edema, ARDS, and fibrosis in severe COVID-19 cases. (**B**). Infection of lungs by *Mycobacterium tuberculosis* (Mtb) causes either symptomatic cavitary progressive tuberculosis (active TB) in individuals with dampened protective immunity or non-progressive LTBI (blue lines) in those with strong immune control. LTBI cases can reactivate to symptomatic TB upon immune-suppressing host conditions. The SARS-CoV-2 infection would happen either in the context of active TB or LTBI (no symptoms). In the SARS-CoV-2 and Mtb co-infected cases (green lines), the host protective immunity is further dampened, leading to exacerbated disease among active TB cases. Besides, SARS-CoV-2 infection of LTBI cases may result in reactivation to active TB (green lines). Corticosteroid (CST) therapy can potentially cause immune-suppression (red line), which augments the dampened protective immunity (purple line), accelerating disease pathology in active TB cases (green line) and reactivates LTBI into symptomatic TB (purple line). Mtb-mediated disease pathologies can be further worsened by SARS-CoV-2 infection; thus, both TB and SARS-CoV-2 promote the disease caused by each of these pathogens.

**Table 1 ijms-22-03773-t001:** Summary of COVID-19 therapy recommended by the United States–National Institute of Health (US-NIH) and US-Center for Disease Control (US-CDC). Data updated on 12 March 2021. ECMO—extracorporeal membrane oxygenation.

Disease Severity	Hospitalization	Medical Support	Treatment Guideline
Mild to moderate	No	No	Anti-SARS-CoV2 antibodies, Bamlanivimab and Etesevimab, are recommended as emergency use authorization for high-risk cases; dexamethasone should not be given.
Mild to moderate	Yes	No	Dexamethasone should not be given. Remdesivir can be used for high-risk cases
Severe	Yes	Oxygen support but not high flow device or ventilation	Remdesivir (alone for minimal oxygen support) and/or dexamethasone can be used. Tocilizumab infusion with dexamethasone therapy.
Critical	Yes	Oxygen support with high flow device or ventilation	Remdesivir and/or dexamethasone can be used. Tocilizumab with dexamethasone therapy.
Critical	Yes	Invasive ventilation or ECMO	Dexamethasone can be used.

**Table 2 ijms-22-03773-t002:** Summary of commonly used corticosteroids and the frequent clinical conditions necessitating their use.

Type and Route of Administration	Name of the Corticosteroid	Clinical Condition
Systemic-Oral, IM, IV	Dexamethasone Prednisone Prednisolone Methylprednisone Hydrocortisone	Inflammatory disorders asthma, COPD, skin rheumatic diseases, brain swelling inflammatory disorders, meningitis, pericarditis, allergies, autoimmune disorders, cancers, adrenal insufficiency
Local-Eye drops, topical or IA	Dexamethasone Prednisolone Methylprednisone Hydrocortisone Medrysone	Eye inflammation, rheumatoid arthritis, skin and connective tissue diseases, allergies, dermatitis
Inhaled along with LABA or SABA or alone	Beclomethasone, Budesonide, Triamcinolone, Fluticasone Flunisolide Ciclesonide Mometasone	Asthma, COPD, bronchitis, pneumonia, allergies

IM—intramuscular; IV—intravenous; IA—intra-articular; COPD—chronic obstructive pulmonary disease; LABA—long-acting beta-agonist; SABA—short-acting beta-agonist.

**Table 3 ijms-22-03773-t003:** Summary of currently ongoing clinical trials that evaluate the usefulness of corticosteroids for the treatment of COVID-19 *.

S.No	Trial ID	Corticosteroids	Criteria	Number of Subjects	Phase	Route of Administration	Study Sponsor	Status	Remarks/Findings
1	NCT04654416	Dexamethasone + Colchicine	Older than 18 years and hospitalized for Covid-19 pneumonia	301	NA	IV	Colombia	Completed	Results awaited
2	NCT04484493	Mometasone furoate nasal spray	18 years or older patients recovered from COVID-19	100	3	Nasal spray	Egypt	Completed	Results awaited
3	NCT04551781	Prednisone	18 years or older patients recovered from COVID-19	450	NA	NA	Egypt	Completed	Results awaited
4	NCT04374071	Methylprednisolone	18 years of age or older hospitalized confirmed COVID-19	250	NA	IV	United States	Completed	Results awaited
5	NCT04273321	Methylprednisolone	18 years of age or older hospitalized confirmed COVID-19	86	NA	IV	China	Completed	Results awaited
6	NCT04445506	Dexamethasone	18 years of age or older hospitalized confirmed COVID-1930% increased CRP	50	NA	NA	United States	Completed	Results awaited
7	NCT04730323	Tocilizumab; methylprednisolone	16 years to 85 years - immunological parameter indicating cytokine storm	93	4	NA	Pakistan	Completed	Results awaited
8	NCT04382053	DFV890	18 years to 80 years hospitalized confirmed COVID-19	143	2	NA	Argentina, Brazil, Denmark, Germany, Hungary, India, Mexico, Netherlands, Peru, Russia, South Africa, Spain	Completed	Results awaited
9	NCT04603729	Dexamethasone; methylprednisone + Tocilizumab	18 years to 75 years hospitalized confirmed COVID-19	100	3	IV	Pakistan	Completed	Results awaited
10	NCT04530409	Early and late Dexamethasone	18 years and older	450	4	NA	Egypt	Recruited	N/A
11	NCT04451174	Prednisone	18 years of age or older hospitalized confirmed COVID-19	184	3		Chile	Recruited	N/A
12	NCT04355247	MethylPREDNISolone	18 years of age or older hospitalized confirmed COVID-19	20	2	IV	Puerto Rico	Recruited	N/A
13	NCT04329650	Siltuximab; methylprednisolone	18 years of age or older hospitalized confirmed COVID-19	200	2	IV	Spain	Recruited	N/A
14	NCT04657484	Medium dose prednisolone; Low dose prednisolone	18 years of age or older hospitalized confirmed COVID-19	100	NA	NA	India	Recruited	N/A
15	NCT04726098	Medium dose prednisolone; Low dose prednisolone	18 years of age or older hospitalized confirmed COVID-19	198	4	NA	Spain	Recruited	N/A
16	NCT04395105	Dexamethasone (high dose)	18 years of age or older hospitalized confirmed COVID-19	284	3	IV	Argentina	Recruited	N/A
17	NCT04355637	Inhaled budesonide	18 years of age or older hospitalized confirmed COVID-19	300	4	Inhaled	Spain	Recruited	N/A
18	NCT04528329	Early; late Dexamethasone	18 years of age or older hospitalized confirmed COVID-19	300	4	Oral; IV	Egypt	Recruited	N/A
19	NCT04486521	IL6 antagonist + corticosteroids	18 years of age or older hospitalized confirmed COVID-19	11000	NA	NA	Saudi Arabia	Recruited	N/A
20	NCT04619693	Dexamethasone	18 years of age or older hospitalized confirmed COVID-19	100	NA	IV	France	Recruited	N/A
21	NCT04559113	Methylprednisone	18 years of age or older hospitalized confirmed COVID-19	200	NA	IV	Pakistan	Recruited	N/A
22	NCT03852537	Methylprednisolone	18 years of age or older hospitalized confirmed COVID-19	90	2	IV	United States,	Recruited	N/A
23	NCT04509973	Dexamethasone	18 years of age or older hospitalized confirmed COVID-19	1000	3	IV	Copenhagen; Denmark; Australia	Recruited	N/A
24	NCT04663555	Dexamethasone	18 years of age or older hospitalized confirmed COVID-19	300	4	IV	Czech Republic	Recruited	N/A
25	NCT04381364	Ciclesonide Inhalation Aerosol	18 years of age or older hospitalized confirmed COVID-19	446	2	Inhaled	Sweden	Recruited	N/A

* As per the US-NIH clinical trial website, accessed on 2 March 2021 [53].

**Table 4 ijms-22-03773-t004:** Clinical parameters of COVID-19 and tuberculosis (TB).

Parameter	COVID-19	TB
Duration of infection	Acute	Chronic
Microorganism	SARS-CoV2	*Mycobacterium tuberculosis*
Primary organ affected	Respiratory system (primarily lungs)	Primarily lungs but any part of the body can be affected
Clinical Symptoms	Fever, cough, sputum production, dyspnea progressing to pneumonia and acute respiratory distress syndrome	Fever, persistent cough, sputum production, malaise, weight loss, night sweats, loss of appetite and
Hosts affected	Animals primarily bats and Humans	Humans
Age group	Active disease in old age but rare in children	Active disease common in all age groups (children, adults and old age)
Comorbidities	Obesity, Hypertension, Diabetes, HIV, cardiovascular patients, respiratory disorders, organ failure patients	Diabetes, HIV, COPD, immunosuppression
Disease severity	Mild, moderate, severe, critical infections	Active, persisters (latency), incipient and subclinical infections
Chest radiology impressions	Bilateral ground-glass opacities	Infiltrates/cavitation/consolidation
Opportunistic Secondary infections	Very common; *Acinetobacter baumannii* and *Klebsiella pneumoniae*	Uncommon except ventilator-associated pneumonia; *Pseudomonas aeruginosa* and *Staphylococcus aureus*

## Data Availability

Not applicable.

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
