# Peer review of "Corticosteroids for COVID-19 Therapy: Potential Implications on Tuberculosis"

_ijms, 2021, doi:10.3390/ijms22073773_

Round 1
Reviewer 1 Report
I have Reviewed the manuscript by Gopalaswamy and Subbian on corticosteroid for COVID-19.
Despite the Abstract being excellent in presenting the scenario for this issue, the first part of the Introduction dwells into aspects of SARS-CoV-2 infection that are not really relevant to the focus of the review. Same with the mention of some other therapies for COVID-19. When they finally get to corticosteroids, they just mention dexamethasone, without providing anything else, no details no other CSTs.
The more important piece that is totally obviated is that CSTs can also be used as adjuvant therapy in TB, were active disease is associated with excessive inflammation (see for example: https://www.ncbi.nlm.nih.gov/pmc/articles/PMC6293474/ ).
Instead they make a huge long unnecessary paragraph on the history of CSTs, and mechanisms of CSTs, when they should be addressing the points I mentioned above.
Concentrate on CSTs in COVID, remove entries 2 and 3, and start with 4. Table 3 needs some serious formatting, and they don’t need to include long names, just country. Add a column for major findings
Concentrate on CST in TB, not viral pneumonia. Is this about TB or viral pneumonia? Make up your mind, and change the title if necessary.
I would discourage the use of informal terms such as whammy.
Rather than making a useless section 6 with Table 4, they should search for case reports of COVID-19 in TB patients.
Author Response
Comment: Despite the Abstract being excellent in presenting the scenario for this issue, the first part of the Introduction dwells into aspects of SARS-CoV-2 infection that are not really relevant to the focus of the review. Same with the mention of some other therapies for COVID-19. When they finally get to corticosteroids, they just mention dexamethasone, without providing anything else, no details no other CSTs.
Response: We thank the reviewer for this suggestion. In the revised version, we have removed some portions which dwell into SARS-CoV2 infection and abridged as “Following aerosol exposure, SARS-CoV-2, the causative agent of COVID-19, usually progresses into the upper and conducting airway from the nasal cavity and can develop symptoms of infection, including rhinitis, sore throat, and sneezing or remain asymptomatic, depending on the host innate immune response. COVID-19 is diagnosed by several molecular and immunological methods across the world [4].” We have also included information on use of other CSTs as follows “Among the CSTs, dexamethasone, administered at 6 mg daily dose (oral or IV) is the preferred choice to treat severe COVID-19 cases. During non-availability of dexamethasone, the United States – National Institute of Health (US-NIH) recommends use of other CSTs at equivalent dose, including prednisone 40 mg; methylprednisone 32 mg and hydrocortisone 160 mg. The half-life, period of activity and frequency of administration should be carefully considered where alternate CSTs are used”. (Page 1-3).
Comment: The more important piece that is totally obviated is that CSTs can also be used as adjuvant therapy in TB, were active disease is associated with excessive inflammation (see for example: https://www.ncbi.nlm.nih.gov/pmc/articles/PMC6293474/ ). Instead they make a huge long unnecessary paragraph on the history of CSTs, and mechanisms of CSTs, when they should be addressing the points I mentioned above. Concentrate on CSTs in COVID, remove entries 2 and 3, and start with 4.
Response: As suggested by the reviewer, we have added the following paragraph on CST as adjuvant therapy for TB: “In combination with standard anti-TB drugs, dexamethasone administered during the first 6 to 8 weeks of treatment has been shown to improve the survival of TB meningitis patients over 14 years of age [81]. Similarly, an overall reduction in relative risk of 22% and 35% has been reported for patients with meningitis and pericarditis, respectively, when corticosteroid was given as adjuvant therapy, along with standard anti-TB drugs [82]. A systematic review on the use of CST to improve the clinical outcome of TB showed a significant beneficial effect of CST therapy, mainly for TB meningitis cases. In contrast, a probable short-term beneficial effect or no effect was reported for patients with pulmonary TB and TB pericarditis, respectively. Adjuvant CST indicated symptomatic relief for TB lymphadenitis and peritonitis but not TB pleuritis [83]. With no further contraindications, dexamethasone or prednisolone administration is recommended for both children and adults as part of TB meningitis therapy for four weeks [84]. A meta-analysis on the use of steroids for abdominal TB showed that the inclusion of steroids could provide some clinical improvement in treated patients [85]. (Page-15).
We have also removed major portions of entry 2, including the following: “Philip Hench, Edward Kendall, and Tadeus Reichstein received the Nobel Prize in medicine and physiology in 1950 for the isolation of the CST, cortisone, which was used to treat rheumatoid arthritis to reduce inflammation [26]” and “The oral or IV CST includes dexamethasone, prednisone, methylprednisolone, and hydrocortisone and is used for acute inflammatory conditions like inflammatory disorders, asthma, chronic obstructive pulmonary disorder (COPD), rheumatic diseases, osteoarthritis, allergies, autoimmune disorders, etc. and given in oral, intramuscular and intrave-nous routes. The inhaled CSTs (ICS) are synthetic steroid compounds like beclomethasone, beclomethasone HFA, budesonide, triamcinolone, ciclesonide, fluticasone, flunisolide, and mometasone are used for treating chronic inflammation, particularly in the lungs. Topical CSTs also include dexamethasone, prednisolone, methylprednisone, hydrocortisone, and medrysone, which are generally used for inflammatory skin diseases”
However, we feel that it is relevant and interesting to include the mechanism of corticosteroid function (entry 3) in the to understand basic molecular pharmacology and wish to retain the same for the benefit of a broader audience.
Comment: Table 3 needs some serious formatting, and they don’t need to include long names, just country. Add a column for major findings.
Response: We thank the reviewer for the suggestion. We revised the table accordingly, and now we have retained only the country names (Page 12-15).
Since most clinical studies have been consolidating their findings, while other studies are still recruiting subjects, there are no results available for any of the study as yet. Therefore, we could not prepare a separate column for major findings.
Comment: Concentrate on CST in TB, not viral pneumonia. Is this about TB or viral pneumonia? Make up your mind, and change the title if necessary.
Response: We agree with this suggestion and removed the section on CST and viral pneumonia.
Comment: I would discourage the use of informal terms such as whammy.
Response: We appreciate the reviewer’s view and we have changed the word “whammy” to “debacle”. (Page-12).
Comment: Rather than making a useless section 6 with Table 4, they should search for case reports of COVID-19 in TB patients.
Response: As suggested by the reviewer, we have added additional case reports on COVID-19 and TB. The following has been added: “Furthermore, four cases of co-existence of COVID-19 and TB was reported in a hospital. In these cases, atypical chest radiographs prompted timely TB diagnosis, since the clinical presentation of cases was otherwise similar to COVID-19 [73]” in the section 6. We kept Table 4 in the best interest of readers to compare the common clinical parameters between COVID-19 and TB. (Page14-15).
Reviewer 2 Report
In this review article, the authors discuss the use of corticosteroids (CST) for COVID-19 therapy, as well as their use for other diseases. They highlight the potential implications for tuberculosis (TB), in particular the problem of reactivating the disease from a latent TB background because of the CST usage. This stresses the importance of screening for underlying TB conditions before the use of CST. The review is well written and addresses a relevant problem in the context of the current SARS-CoV2 pandemic. I have noted some minor points that require revisions before submission:
I would recommend following revisions to Figure 1: group the gene names, GR-responsive elements and cellular responses in boxes, with a bigger font (barely visible now); simplify the scheme by removing arrows and keeping only activator/inhibitor arrows going from box to box; Increase the overall size of the font for the element of importance (e.g. extracellular space, nucleus…).
The correct name is Remdes-i-vir. Please correct everywhere it applies.
There is an unfinished sentence in the introduction: “A few studies indicated the”
Beginning of Part 4. “In the randomized […]”
Table 3, entry 24: there are “#” characters displayed, probably in place of special characters
Table 4, for disease severity for TB, “Resisters” is mentioned twice
Part 6. The second paragraph about TB granuloma would require some revisions: Upon primary infection, the mycobacteria keeps macrophages under control and prevent the “cytokine storm” from happening, which is why primo-infection is mostly asymptomatic even in immuno-competent host. Symptoms occur when the lung are damaged through cavitation or fibrocaseation. The mechanisms leading to the formation of these lesions are not fully understood, but usually involve sudden leakage of otherwise confined, bacteria-loaded, necrotic materials in healthy tissues. This can happen upon physical disruption of non-calcified granuloma or obstructed bronchioles filled with caseum, upon coughing or sneezing for example. Failure to generate the granuloma (immunocompromised host) lead to disseminated disease but absence of lung cavities. Suggested readings and citations:
- Hunter RL. The pathogenesis of tuberculosis: the early infiltrate of post-primary (adult pulmonary) tuberculosis: a distinct disease entity. Frontiers in immunology. 2018 Sep 19;9:2108.
- Ihms EA, Urbanowski ME, Bishai WR. Diverse cavity types and evidence that mechanical action on the necrotic granuloma drives tuberculous cavitation. The American journal of pathology. 2018 Jul 1;188(7):1666-75.
Figure 2, panel B. The figure should be updated to better reflect the objective of the review: The SARS-CoV2 infection would happen in the context of a TB-infected host, either presenting active TB and associated symptoms, in which case co-infection worsen the outcome, or presenting no symptoms (latent TB), in which case the use of CST can prevent efficient bacterial control in the granuloma and lead to disease reactivation.
Conclusion: What’s about the use of anti-TB drugs in parallel to CST for severe COVID-19 with latent TB diagnosis? This could prevent TB from reactivating while keeping the beneficial effects of CST on COVID, but this strategy is not discussed anywhere in the review. It would be appropriate if the authors could comment on this or at least leave it open for future consideration. Whether the TB treatment is effective/feasible in patients presenting such conditions (SARS-CoV2 infection + CST) is an important point to answer.
Author Response
Comment: In this review article, the authors discuss the use of corticosteroids (CST) for COVID-19 therapy, as well as their use for other diseases. They highlight the potential implications for tuberculosis (TB), in particular the problem of reactivating the disease from a latent TB background because of the CST usage. This stresses the importance of screening for underlying TB conditions before the use of CST. The review is well written and addresses a relevant problem in the context of the current SARS-CoV2 pandemic.
Response: We thank the reviewer for the positive feedback and valuable suggestions.
Comment: I would recommend following revisions to Figure 1: group the gene names, GR-responsive elements and cellular responses in boxes, with a bigger font (barely visible now); simplify the scheme by removing arrows and keeping only activator/inhibitor arrows going from box to box; Increase the overall size of the font for the element of importance (e.g. extracellular space, nucleus…).
Response: As recommended by the reviewer, we have completely modified Figure-1 with clearly visible font and all genes and cellular responses grouped in boxes. (Page-3).
Comment: The correct name is Remdes-i-vir. Please correct everywhere it applies.
Response: We thank the reviewer for pointing out this error. We have corrected “remdesivir” appropriately throughout in the revised manuscript.
Comment: There is an unfinished sentence in the introduction: “A few studies indicated the”
Response: We thank the reviewer for pointing out this error. We have removed that incomplete sentence from the introduction.
Comment: Beginning of Part 4. “In the randomized […]”
Response: We thank the reviewer for the suggestion. We have added the same to the beginning of part 4. (Page-8).
Comment: Table 3, entry 24: there are “#” characters displayed, probably in place of special characters
Response: We thank the reviewer for the suggestion. As suggested by the other reviewer, we have retained only the country names on the table and hence the above mentioned character has been suitably removed. (Page 8-12).
Comment: Table 4, for disease severity for TB, “Resisters” is mentioned twice
Response: We thank the reviewer for the suggestion. We have deleted “Resisters” on Table-4 since that word does not fit with the clinical description of TB. (Page-13).
Comment: Part 6. The second paragraph about TB granuloma would require some revisions: Upon primary infection, the mycobacteria keeps macrophages under control and prevent the “cytokine storm” from happening, which is why primo-infection is mostly asymptomatic even in immuno-competent host. Symptoms occur when the lung are damaged through cavitation or fibrocaseation. The mechanisms leading to the formation of these lesions are not fully understood, but usually involve sudden leakage of otherwise confined, bacteria-loaded, necrotic materials in healthy tissues. This can happen upon physical disruption of non-calcified granuloma or obstructed bronchioles filled with caseum, upon coughing or sneezing for example. Failure to generate the granuloma (immunocompromised host) lead to disseminated disease but absence of lung cavities. Suggested readings and citations: Hunter RL. The pathogenesis of tuberculosis: the early infiltrate of post-primary (adult pulmonary) tuberculosis: a distinct disease entity. Frontiers in immunology. 2018 Sep 19;9:2108. Ihms EA, Urbanowski ME, Bishai WR. Diverse cavity types and evidence that mechanical action on the necrotic granuloma drives tuberculous cavitation. The American journal of pathology. 2018 Jul 1;188(7):1666-75.
Response: We thank the reviewer for sharing the thought. As rightly said, the mechanisms underlying granuloma maturation and cavity formation are not fully understood. We have modified this section as follows: “Upon infection of the pulmonary alveoli, Mtb is thought to be phagocytosed by innate immune cells, such as the alveolar macrophages and dendritic cells, which use pattern recognition receptors (PRRs), predominantly toll-like receptors, expressed on their surface to interact with the pathogen-associated molecular patterns (PAMPs) of the bacteria. Following Mtb engulfment, the phagocytes produce a plethora of proinflammatory cytokines, including TNF-α, IL-1β, IL-6, IL-12, IL-18, IL-23, and IFN-γ, and chemokines. Some of these molecules, including IL-6, IL-12, IL-1β and IL-18, also constitute the inflammatory cytokine storm seen in severe COVID-19 cases. In TB, the cytokine/chemokine milieu attracts more immune cells from the circulation to the infection site, forming the granuloma [60,61]. Mtb-infected phagocytes are at the center of granuloma, followed by layers of different types of immune cells, such as monocytes, epithelioid cells, foamy macrophages, multi-nucleated giant cells, and neutrophils. Lymphocytes, including T and B cells, are mostly present as a ring or cuff at the outer side of the granulomas [60,61]. On the other hand, the infecting Mtb deploys strategies to prevent the proinflammatory responses of infected phagocytes, leading to asymptomatic latent infection (LTBI). During reactivation of LTBI, the lungs are damaged through cavitation or fibrocaseation of granulomas, which results in the occurrence of clinical symptoms. The mechanisms leading to the formation of these lesions in TB reactivation are not fully understood, although it might involve sudden leakage of otherwise confined, bacteria-loaded, necrotic materials in healthy tissues [62,63]. This may occur either due to the physical disruption of non-calcified granuloma or obstructed bronchioles filled with caseum. Mtb can proliferate exorbitantly within these cavitary granulomas, facilitating bacterial dissemination from the lungs to other organs [60,61]. Furthermore, the expulsion of phlegm (caseum) upon coughing or sneezing contributes to disseminating Mtb infection between individuals [62,63].” and added the references (62, 63) as indicated by the reviewer. (Page 13-14).
Comment: Figure 2, panel B. The figure should be updated to better reflect the objective of the review: The SARS-CoV2 infection would happen in the context of a TB-infected host, either presenting active TB and associated symptoms, in which case co-infection worsen the outcome, or presenting no symptoms (latent TB), in which case the use of CST can prevent efficient bacterial control in the granuloma and lead to disease reactivation.
Response: As suggested by the reviewer, we have modified Figure 2, panel B and revised the figure legend accordingly. (Page17-18).
Comment: Conclusion: What’s about the use of anti-TB drugs in parallel to CST for severe COVID-19 with latent TB diagnosis? This could prevent TB from reactivating while keeping the beneficial effects of CST on COVID, but this strategy is not discussed anywhere in the review. It would be appropriate if the authors could comment on this or at least leave it open for future consideration. Whether the TB treatment is effective/feasible in patients presenting such conditions (SARS-CoV2 infection + CST) is an important point to answer.
Response: We thank the reviewer for this important suggestion. We have added the following sentences: “Furthermore, a preventive latent TB drug therapy could be recommended for COVID-19 cases identified positive for TB in screening. In fact, the WHO and CDC guidelines recommends tuberculosis preventive therapy (TPT) for latent TB treatment. The TPT regimen includes 3 to 6 months treatment with anti-TB drugs, isoniazid either alone or in com-bination with rifampicin or rifapentine [99,100]. For risk groups, such as CST users, the CDC recommends using prednisone at 15mg/day for at least a month [99]. Clinical trials conducted among the high risk group of non-COVID-19 cases showed 27-95% reduction in incidence of active TB for those on TPT, compared to placebo, which was considered effective [101,102]. It is intriguing to consider prophylactic treatment with anti-TB drugs in parallel to CST for patients diagnosed with severe COVID-19 and LTBI. This treatment strategy could prevent reactivation of LTBI into symptomatic TB, while keeping the beneficial effects of CST in controlling the severity of COVID-19. Whether the TB treatment is effective or even feasible for patients presenting such conditions (SARS-CoV2 infection + CST) is an important point to answer. Additional and extensive clinical studies are needed to address these co-morbid conditions for effective management of both COVID and TB” (Page-17).
Round 2
Reviewer 1 Report
This is certainly an improvement to the initial version.
The following long paragraph needs to be removed, as it does not belong to the focus of the paper and it doesn't add anything substantial:
"At present, the diagnostic tests for TB and COVID-19 are mostly performed
sequentially rather than concurrently, depending on the initial
disease symptoms and prior history of exposure to Mtb. In general, tuberculin
skin testing (TST), interferon-gamma release assay (IGRA)
and/or chest X-ray are used to determine the Mtb exposure status. However,
such screening procedures are complicated by the ongoing COVID-
19 pandemic. For example, studies have shown that mobile X-ray unit
was a cause of nosocomial SARS-CoV-2 infection in a TB screening setting.
In a case report, a patient with COVID-19 developed bloody sputum
along with fever and cough. Upon testing, it was confirmed that the patient
had a latent Mtb infection, which progressed to an active TB disease
in the presence of COVID-19 [69]. This observation indicates that although
it takes a longer time for Mtb infection to progress into active disease,
in the presence of an inflammatory disease condition, such as
COVID-19, the progression of TB is accelerated by the weakened host immunity.
Therefore, individuals exposed/infected with Mtb must be carefully
monitored in situations such as the current COVID-19 pandemic
[70]. In a study involving two patient cohorts spanning 8 countries, a
10.6% fatality rate was reported among 8 out of 69 cases of COVID-19 and
TB where the SARS-CoV-2 infection was nosocomial and was diagnosed
after the confirmation of preliminary TB infection [71]. In another cohort
study that evaluated the consequence of either COVID-19 or TB presenting
as sequelae to one another or co-diagnosed, a 12% fatality rate of patients
was reported when TB and COVID-19 co-existed [72]. Furthermore,
Int. J. Mol. Sci. 2021, 22, x FOR PEER REVIEW 15 of 25
four cases of co-existence of COVID-19 and TB were reported in a hospital.
Atypical chest radiographs prompted timely TB diagnosis in these
cases since the clinical presentation of cases was otherwise similar to
COVID-19 [73]. Thus, when the chest radiology impressions show a complex
disease pathology, combined with pneumonia or acute distress syndrome,
it is important to consider the concurrent diagnosis of TB and
COVID-19 and proper treatment strategy. [74-78]."
Some English editing is still needed. e.g. "...four cases of co-existence of COVID-19 and TB was reported in a hospital. In these cases.."
Author Response
Comment: The following long paragraph needs to be removed, as it does not belong to the focus of the paper and it doesn't add anything substantial:
"At present, the diagnostic tests for TB and COVID-19 are mostly performed sequentially rather than concurrently, depending on the initial disease symptoms and prior history of exposure to Mtb. In general, tuberculin skin testing (TST), interferon-gamma release assay (IGRA) and/or chest X-ray are used to determine the Mtb exposure status. However, such screening procedures are complicated by the ongoing COVID-19 pandemic. For example, studies have shown that mobile X-ray unit was a cause of nosocomial SARS-CoV-2 infection in a TB screening setting. In a case report, a patient with COVID-19 developed bloody sputum along with fever and cough. Upon testing, it was confirmed that the patient had a latent Mtb infection, which progressed to an active TB disease in the presence of COVID-19 [69]. This observation indicates that although it takes a longer time for Mtb infection to progress into active disease, in the presence of an inflammatory disease condition, such as COVID-19, the progression of TB is accelerated by the weakened host immunity. Therefore, individuals exposed/infected with Mtb must be carefully monitored in situations such as the current COVID-19 pandemic [70]. In a study involving two patient cohorts spanning 8 countries, a 10.6% fatality rate was reported among 8 out of 69 cases of COVID-19 and TB where the SARS-CoV-2 infection was nosocomial and was diagnosed after the confirmation of preliminary TB infection [71]. In another cohort study that evaluated the consequence of either COVID-19 or TB presenting as sequelae to one another or co-diagnosed, a 12% fatality rate of patients was reported when TB and COVID-19 co-existed [72]. Furthermore, Int. J. Mol. Sci. 2021, 22, x FOR PEER REVIEW 15 of 25 four cases of co-existence of COVID-19 and TB were reported in a hospital. Atypical chest radiographs prompted timely TB diagnosis in these cases since the clinical presentation of cases was otherwise similar to
COVID-19 [73]. Thus, when the chest radiology impressions show a complex disease pathology, combined with pneumonia or acute distress syndrome, it is important to consider the concurrent diagnosis of TB and COVID-19 and proper treatment strategy. [74-78]."
Response: We appreciate the reviewers’ view. However, we would like mention that the same reviewer had added a comment in the first round of revision that “Rather than making a useless section 6 with Table 4, they should search for case reports of COVID-19 in TB patients”
In our revised version, we have revised the concerned paragraph and discussed the case reports of COVID-19 and TB infections that occurred as a sequela to one another. We still feel that they are very relevant to the topic, since it highlights the implications of the two deadly respiratory diseases on the diagnosis and management of each other.
The revised paragraph now reads “The combined impact of COVID-19 and TB on patient mortality/morbidity and disease management is confounded by several factors, primarily the lack of a strategy to diagnose the cases for both diseases. At present, the diagnostic tests for TB and COVID-19 are mostly performed sequentially rather than concurrently, depending on the initial disease symptoms and prior history of exposure to Mtb. In general, tuberculin skin test (TST), interferon-gamma release assay (IGRA) and/or chest X-ray are used to determine the Mtb exposure status. However, such screening procedures are complicated by the ongoing COVID-19 pandemic. For example, a mobile X-ray unit used for TB screening was found to be a cause of nosocomial SARS-CoV-2 infection. In a case report, a patient with COVID-19 developed bloody sputum along with fever and cough. Upon testing, it was confirmed that the patient had LTBI, which progressed to an active TB disease in the presence of COVID-19 [69, 70]. Similarly, in a clinical study involving patient cohorts from 8 countries, where the SARS-CoV-2 infection was nosocomial, a 10.6% fatality rate was reported among 8 out of 69 confirmed cases of COVID-19 and TB [71]. In another cohort study that evaluated the consequence of either COVID-19 or TB presenting as sequelae to one another or co-diagnosed, a 12% fatality rate of patients was reported when TB and COVID-19 co-existed [72]. Furthermore, four cases of COVID-19 and TB were reported in a hospital, with clinical presentation indistinguishable between these two diseases. However, atypical chest radiograph findings prompted timely TB diagnosis in these cases [73]. These studies indicate that although it takes a longer time for healthy individuals with LTBI to progress into active disease, co-infection with SARS-CoV-2 accelerates the progression of TB by weakening the host immunity. Hence, when the chest radiology impressions show a complex disease pathology, combined with pneumonia or acute distress syndrome, it is important to consider the concurrent diagnosis of TB and COVID-19 and proper treatment strategy [74-78].” We hope that the reviewer agrees with our view.
Comment: Some English editing is still needed. e.g. "...four cases of co-existence of COVID-19 and TB was reported in a hospital. In these cases.."
Response: We thank the reviewer and have changed the sentence as “Furthermore, four cases of COVID-19 and TB were reported in a hospital, with clinical presentation indistinguishable between these two diseases. However, atypical chest radiograph findings prompted timely TB diagnosis in these cases”.
Also, we revised the manuscript for more grammar and formatting edits as indicated in the “track change-version”.